# Nutritional Quality, Safety and Environmental Benefits of Alternative Protein Sources—An Overview

**DOI:** 10.3390/nu17071148

**Published:** 2025-03-26

**Authors:** Anna Choręziak, Dawid Rosiejka, Joanna Michałowska, Paweł Bogdański

**Affiliations:** Department of Obesity and Metabolic Disorders Treatment and Clinical Dietetics, Poznań University of Medical Sciences, 60-355 Poznań, Poland

**Keywords:** plant protein, edible insects, water protein source, fungal proteins, alternative protein sources, sustainable diet

## Abstract

Protein is a fundamental macronutrient in the human diet. It supplies our organisms with essential amino acids, which are needed for the growth and maintenance of cells and tissues. Conventional protein sources, despite their complete amino acid profiles and excellent digestibility, have a proven negative impact on the environment. Furthermore, their production poses many ethical challenges. This review aims to present nutritional, more ethical, and environmentally friendly alternatives that could serve as potential protein sources for the population. The available literature on alternative protein sources has been analyzed. Based on the research conducted, various products have been identified and described, including plant-based protein sources such as soybeans, peas, faba beans, lupins, and hemp seeds; aquatic sources such as algae, microalgae, and water lentils; as well as insect-based and microbial protein sources, and cell-cultured meat. Despite numerous advantages, such as a lower environmental impact, higher ethical standards of production, and beneficial nutritional profiles, alternative protein sources are not without limitations. These include lower bioavailability of certain amino acids, the presence of antinutritional compounds, technological challenges, and issues related to consumer acceptance. Nevertheless, with proper dietary composition, optimization of production processes, and further technological advancements, presented alternatives can constitute valuable and sustainable protein sources for the growing global population.

## 1. Introduction

It is estimated that the global population will reach 10 billion in the next 25 years, which will be associated with an estimated increase in food production of 70%. One nutrient that will be in particular short supply in the future is protein [1]. This prediction is consistent with statements by the Food and Agriculture Organization of the United Nations (FAO), which estimates that by 2050, global meat production will double from its current level. This fact highlights the increasing need to search for new food sources and alternative food systems [2]. Moreover, a large part of the Earth’s land resources is used for the production of animal food, which contributes to environmental problems such as the degradation of natural habitats, threats to biodiversity, production of greenhouse gases, soil degradation, water pollution, overgrazing and desertification, and thus to the ecological crisis and climate change. It is necessary to search for other sources of protein and methods of production to meet consumer demands and reach the estimated global protein needs. Protein is a key macronutrient in the human diet. This nutritional compound is the basic structural and functional component of every cell in the human body. It is essential for the development and growth processes of young organisms. Proteins are responsible for the possibility of selective transport from and to cells, enabling chemical reactions and direct metabolic pathways. After release in the digestive tract, peptides derived from dietary proteins can affect digestive processes (secretion and transport) or modulate the absorption of nutrients. Proteins are substrates in the synthesis of many hormones and biologically active compounds, such as adrenaline and noradrenaline, thyroid hormones (thyroxine and triiodothyronine), serotonin, and histamine. Insufficient protein intake causes reduced muscle mass and muscle weakness, as well as changes in hair and skin [3]. Dairy products and meat are among the most commonly consumed and nutritionally balanced animal proteins. However, the increase in demand is not optimal for the environment due to the low efficiency and large environmental footprint created during the production process [1]. It is necessary to find more sustainable methods of producing animal proteins and to balance their share with plant proteins or their alternatives. This approach provides the development of more sustainable and environmentally friendly food systems, supports biodiversity and ensures efficient distribution of good-quality proteins to the entire population [4,5]. Increasing consumer awareness, political pressure, and the global context to include more plant proteins in people’s diets contribute to the discussion about the purpose of alternative protein sources and their impact on the human body [6]. Examples of currently examined and sustainable proteins include legumes (e.g., soybeans, peas, beans), seaweeds and algae, fungi (mycoproteins), hemp seeds, insects, and cell-based meat. Although the search for new sources of protein is becoming inevitable, the reconciliation of food needs cannot be based solely on nutritional facto but also on sustainability [7]. The aim of this paper is to present the current knowledge on alternative sources of proteins, their nutritional value, their potential use in the food industry and their impact on human health.

## 2. Literature Search Methodology

To gather the relevant literature for this review, we conducted a structured search in four major scientific databases: PubMed, Scopus, ScienceDirect, and Google Scholar. The search focused on peer-reviewed articles published in English, prioritizing recent studies that contribute to the current understanding of alternative protein sources.

We used a combination of keywords and Boolean operators to refine the search results. The primary keywords included: “soybean”, “pea”, “faba bean”, “lupin”, “hemp seeds” (including their Latin names), “water lentils” or “duckweed”, “algae”, “macroalgae”, “microalgae”, “*Spirulina*”, “*Chlorella*”, “insect protein”, “edible insect”, “entomophagy”, “microbial protein”, “fungal protein”, “cultured meat”, “cellular agriculture”, and “cultivated meat”, along with variations and related terms as necessary. Publications issued after 2000 were considered for inclusion. Articles were selected based on their relevance to the research focus, prioritizing studies that provided empirical data, theoretical insights, or significant contributions to the field. While this is not a systematic review, our approach aims to provide a comprehensive and representative overview of the existing literature.

## 3. Plant-Based Protein Sources

### 3.1. Legume Seeds

Legume seeds have been a part of the human diet for thousands of years. They are often used in folk medicine as products with versatile healing properties. These annual plants belong to the legume family, and their edible part is the fruit, i.e., the pod, which can be eaten in whole or in part, depending on the maturity and type of fruit. Edible legume species include beans, peas, peanuts, chickpeas, lentils, broad beans and soybeans [8]. Legumes are characterized by high nutritional value. They are a very good source of protein with high biological value, unsaturated fatty acids, fiber, vitamin B, minerals such as magnesium, iron, calcium and potassium, and bioactive phytochemicals such as flavonoids, phenolic acids, carotenoids, phytosterols [9,10]. Moreover, they have a low glycemic index and contain α-amylase and α-glucosidase inhibitors, which regulate glucose absorption from the digestive tract [8]. The percentage of protein in legume seeds ranges from 17 to 20% (dry weight) in peas and beans to 38 to 40% in soybeans, depending on the species. This is similar to the protein content in meat, which usually consists of 18–20% of this macronutrient [11]. The fat content in legumes ranges from 2 to 21%, depending on the type and species. They have a high concentration of triglycerides monounsaturated and polyunsaturated fatty acids, such as linoleic and linolenic acid. Most legume seeds contain a small amount of saturated fatty acids, less than 1% of the total fat content [12]. Carbohydrates constitute 6–62% of the weight of legume seeds, depending on the species and genus. This group includes starch, which, after processing, changes into resistant starch. In this way, it acquires its functional properties. Other carbohydrates present in legumes are monosaccharides, disaccharides and galactosides [11]. Legume seeds are a very good source of dietary fiber, which includes resistant starch, non-starch polysaccharides (cellulose, hemicellulose, pectin, gums and β-glucans), indigestible oligosaccharides and lignin [12]. The ratio of soluble to insoluble fiber in legume seeds is comparable to that in grains. The fiber content, depending on the species, variety and seed processing, can range from 8% to 27.5%, and approximately 3% to 14% of this concentration corresponds to soluble fiber [8,11,12]. Legumes are also an important source of minerals, containing significant amounts of iron, zinc, calcium, and magnesium. The content of these nutrients varies depending on the species and variety of seeds. However, the concentrations of iron, zinc, and calcium are lower and less absorbable than those of animal origin [10,12]. Moreover, legume sprouts, especially soybean, alfalfa or mung bean sprouts, are also worth considering. They are gaining more and more attention as functional foods due to their nutritional values, including amino acids, fiber, trace elements; copper, manganese and selenium, vitamins C and E, flavonoids (apigenin, kaempferol, myricetin, naringin, quercetin, rutin, daidzein and genistein) and phenolic acids (gallic and caffeic acid) [13,14,15]. Due to the nutritional profile and wide range of potential health benefits of consuming legumes, dietary guidelines from several organizations recommend their regular consumption [16,17]. FAO indicates that legumes are an important part of the diet with significant benefits for human and planetary health [18]. It has been shown that consuming 50 g of legumes per day reduces the risk of all-cause mortality [19]. Moreover, several studies demonstrated that legume consumption is associated with a lower risk of cardiovascular disease (CVD) and selected CVD-related factors, such as obesity, high blood pressure, dyslipidemia, and type 2 diabetes (T2D) [9,20,21,22]. Soy consumption is associated with a reduced risk of several cancers, including breast and prostate cancer [23]. Population studies show that soy consumption is associated with breast cancer prevention and recurrence risk in people who have previously had breast cancer [20,24]. In addition, legume consumption contributes to reducing oxidative stress, reducing free radical damage, promoting gut microbiota diversity in gut health, and inhibiting inflammation [25,26]. Furthermore, legumes have a low glycemic index and high dietary fiber content, which is important in the prevention and treatment of diabetes. In addition, they contain α-amylase and α-glucosidase inhibitors, which contribute to reduced glucose absorption from the gastrointestinal tract [8]. Besides the various health benefits of legume consumption, legumes also play a role in sustainable food production and fit well into nutritious and environmentally friendly dietary patterns [27,28]. Their consumption may also have a socioeconomic significance through their contribution to reduced healthcare costs and the prevention of cardiometabolic diseases [29,30].

The most well-known and consumed legume seeds include soybeans, chickpeas, lentils, beans, peas, broad beans, and peanuts. Below, we describe in detail selected types of leguminous plants that can be considered great alternative protein sources.

#### 3.1.1. Soybean

Soybean (*Glycine max*) is an economically significant food and oilseed crop, serving as a major source of plant-based protein and oils. This nutrient-rich legume has been a staple in traditional Asian diets for thousands of years and is now gaining popularity in Western countries as a versatile protein source used in a wide range of food products [31].

Soy protein is considered a complete protein as it contains all essential amino acids necessary for human nutrition. It is particularly high in lysine, whereas methionine and cysteine can be considered as limiting amino acids [32,33]. High digestibility and quality comparable to animal proteins make it a suitable alternative for vegetarians and those allergic to milk protein [34,35]. Soybeans are rich in essential nutrients, including high-quality vegetable protein, soluble fibers, oligosaccharides, minerals, vitamin B, soy lecithin, and the bioactive phytoestrogens, particularly the isoflavones genistein and daidzein. They are low in saturated fat and a source of unsaturated and omega-3 (*n*-3) fatty acids [36,37].

Soybeans undergo several processing steps, starting with cleaning, drying, and husking to remove the shell, which is used for fiber additives. The beans are then rolled into flakes for animal feed or further processed into flour, oil, or protein products. Defatted soy flakes serve as the base for soybean powder, concentrates, and isolates, which are widely used in food production. Soy-based foods fall into unfermented (e.g., tofu, soymilk) and fermented categories (e.g., miso, tempeh, soy sauce), offering diverse nutritional and culinary applications [38].

Soy protein consumption is associated with lower levels of total cholesterol, low-density lipoproteins, and triglycerides, contributing to improved cardiovascular health [39,40,41]. Fermented soy products offer various health benefits, particularly for individuals with T2D. Miso and tempeh possess antioxidant properties and help reduce insulin resistance. Natto (fermented soybeans) not only has antioxidant and anti-inflammatory effects but also inhibits glucose uptake in intestinal cells and exhibits dipeptidyl peptidase IV (DPP-IV) inhibitory activity. Fermented soy milk shares similar antioxidant and anti-inflammatory benefits while also modulating GLUT4 expression in muscles, which aids glucose uptake and metabolism [42]. Soy isoflavones support bone health by reducing bone resorption and promoting bone formation, helping to maintain overall bone density. While some studies show promising effects on bone metabolism, more research is needed to confirm their role in preventing bone disorders [43]. Diets high in soy products consumption, especially soymilk and tofu, are associated with lower cancer risk, particularly gastrointestinal and gynecological cancers. More prospective cohort studies are still needed to confirm the causal relationship between soy product consumption and cancer risk [44]. Soy isoflavones may help alleviate menopause symptoms by improving quality of life, including vasomotor, psychosexual, physical, and sexual well-being. A study found that postmenopausal women taking high-dose isoflavones experienced significant symptom relief compared to a placebo group. These findings suggest that soy isoflavones could be a safe and effective alternative to estrogen therapy for managing menopause symptoms [45]. Concerns have been raised regarding the potential negative impact of soy intake on thyroid function, particularly in individuals with iodine deficiency or subclinical hypothyroidism. However, when patients receive appropriate treatment, such as levothyroxine, along with iodine supplementation, a balanced diet, and adequate micronutrient support, thyroid dysfunction—like soymilk-induced hypothyroidism and goiter—can be effectively reversed. This highlights the importance of careful dietary assessments and supplementation to prevent nutritional deficiencies, especially in children with restrictive diets due to multiple food allergies [46,47].

Soybeans are a cost-effective and abundant source of protein, making them a viable option for feeding larger populations, especially in regions with protein deficiencies. Their versatility in food production allows for the creation of meat analogs and milk substitutes, which can help to meet the nutritional needs of growing populations. By fortifying soy beverages with calcium, the nutritional gap with cow’s milk is successfully bridged, making fortified soy drinks the preferred choice among consumers seeking a wholesome alternative to traditional dairy products [48,49]. The study proved the possibility of considerably improving human health in Se and Zn deficient regions through Se/Zn food and forage enrichment in forms easily absorbable by the mammal body. This can be achieved by biofortification of soybean seeds with Se and Zn spiked to Se-deficient soil during the flowering phase [49].

Soy cultivation has significant environmental implications, including land use, water consumption, and CO_2_ emissions. However, compared to animal protein sources, soybeans are generally more sustainable, requiring less land and water while producing lower greenhouse gas emissions [50]. Around 30 million tons go toward animal feed, while only 13 million tons—approximately 2.5 times less—are used directly for human consumption in foods like tofu, soy milk, and tempeh [51].

Recent studies have highlighted the significant interest in soybeans among researchers due to their potential health benefits. For instance, a comprehensive review demonstrated that soybean consumption is inversely associated with risks of various cancers, type 2 diabetes mellitus, and menopausal symptoms [52]. Additionally, advancements in processing techniques have enhanced the nutritional profile of soy proteins, making them valuable in both food and pharmaceutical applications [53,54]. These findings underscore the importance of soybeans in promoting health and preventing diseases.

#### 3.1.2. Pea

Pea (*Pisum sativum*) belongs to the legume family and is one of the most widely cultivated legumes in the world. It accounts for 26% of the total production of legume seeds. The nutritional value of pea protein depends on the variety, growth conditions and protein extraction methods used to produce isolates and concentrates [55]. Pea seeds usually contain about 20–25% protein, 40–50% carbohydrates, mainly starch, and 10–20% dietary fiber, but also about 1.5–2% fat. Pea proteins mainly consist of four groups: globulins (65–80%), albumins (10–20%), prolamins and glutelins [56]. Moreover, peas are a source of vitamins, mainly B vitamins, minerals, mainly potassium, phosphorus, magnesium and calcium, and polyphenols. They also contain essential amino acids with a high content of lysine and threonine, while sulfur-containing amino acids, i.e., methionine and cysteine, are deficient [57,58,59]. It should be noted that antinutritional substances, such as protease inhibitors, phytic acid, oxalates and tannins, are also present [60].

Eating products rich in pea protein helps reduce the risk of cardiovascular diseases and diabetes. Its consumption has also been associated with a protective effect against various cancers (such as breast, colon and kidney cancer) [61]. Pea protein is hypoallergenic and has antioxidant, antihypertensive, anti-inflammatory and cholesterol-lowering effects [57,62]. It is a source of bioactive peptides that, along with antioxidant properties, show an inhibitory effect on the angiotensin I-converting enzyme (ACE), which contributes to health improvement [56]. Eating peas has an appetite-suppressing effect, which may result from the high content of protein, starch and dietary fiber. These components contribute to delayed gastric emptying, reduced and slower glucose absorption, and stimulation of the release of appetite-regulating hormones [57,61].

Pea proteins are becoming increasingly popular and more widely used in the food industry due to their wide availability, nutritional value, cost-effectiveness, and health benefits. They are considered functional ingredients that increase the protein content of the diet while fulfilling a certain function, such as thickening agents, gelling agents, emulsion and foam stabilizers, and fat binders. Pea protein can be used in the production of cereal, bakery, meat, and dairy products [56,57]. The properties of its fractions to form soft gels enable the development and production of alternative beverages to dairy, fermented, and cottage cheese products [62]. However, the industrial use of pea proteins is partially limited due to the content of antinutritional components, an undesirable beany aftertaste, and a complex spherical structure. The introduction of protein modifications can affect these properties and increase the degree of use of pea proteins in the commercial industry. These changes can be achieved by enzymatic, chemical and physical modifications. Reaction conditions, modifiers and the extent of modification are the most important factors for protein alterations. Their proper control allows them to obtain the desired properties that can be used in the production of meat analogs, cereal products, dairy product imitations, capsules and emulsions. It is necessary to understand the characteristics of pea proteins in order to design appropriate solutions and better use them in the food industry [57,62,63].

#### 3.1.3. Fava Bean

*Vicia faba* L., commonly known as fava bean or broad bean, is a legume from the Fabaceae family that is cultivated and consumed worldwide. It offers significant nutritional and environmental benefits, serving as a rich source of plant-based protein and essential amino acids, making it a valuable crop for both human and animal nutrition [64].

Faba bean seeds are rich in protein, with a content of approximately 20–35% on a dry matter basis. The major proteins are globulin storage proteins, enzymes, and lectins. In the faba bean seed, proteins are mostly stored in cotyledon, with globulins, albumin, prolamin, and glutenin being the most abundant, accounting for nearly 80% of the total seed protein in weight [65,66]. Compared to cereals, dry faba bean seeds are rich in lysine but have lower levels of sulfur-containing amino acids such as methionine, cysteine, and tryptophan. In contrast, most cereal grains are low in lysine but high in these sulfur-containing amino acids. This complementary amino acid profile makes faba beans and cereals ideal for pairing in a plant-based diet, ensuring a more balanced and complete protein intake [67,68]. The lipid content of broad beans is relatively low, typically ranging from 0.70% to 2.00% of their dry weight. The high-starch fraction of faba beans is a valuable component, and they are also rich in bioactive compounds, particularly in polyphenols and flavonoids, which exhibit strong antioxidant properties. Additionally, they are a good source of essential minerals such as iron, phosphorus, magnesium, and potassium. They also contain significant amounts of B vitamins, including folate and niacin [67,69].

Faba bean seeds can be processed into various ingredients, such as flour, protein concentrate, and protein isolate, which are potential ingredients for the production of meat analogs [70]. Various technological techniques have been proposed to reduce antinutrients in fava beans and enhance their utilization in the food industry. Antinutritional factors such as lectins, saponins, trypsin inhibitors, phytic acid, and condensed tannins can be reduced through non-thermal methods (dehulling and soaking), thermal methods (cooking, autoclaving, microwaving, and extrusion), and biological methods (germination and fermentation) [71,72,73].

Faba beans offer numerous health benefits due to their rich composition of bioactive compounds. Their bioactive peptides have immune-modulating properties and contribute to lowering low-density lipoprotein (LDL) cholesterol levels, as well as reducing hepatic cholesterol and triacylglycerol, without affecting high-density lipoprotein (HDL) cholesterol. Additionally, faba bean oligosaccharides promote the growth of beneficial gut microbiota, enhance gut signaling and metabolism, and help maintain a healthy gut barrier. These effects not only support digestive health but also reduce chronic inflammation, lowering the risk of colon cancer. Furthermore, faba beans may help regulate blood sugar levels, making them a valuable addition to a balanced diet [64,66,74].

The presence of antinutritional factors such as vicine and convicine can trigger favism in individuals with glucose-6-phosphate dehydrogenase (G6PD) deficiency and reduce feeding efficiency in monogastric animals. G6PD deficiency, a common sex-linked erythrocyte enzyme disorder, is caused by mutations in the Gd gene. Favism refers to an acute hemolytic episode that occurs in G6PD-deficient individuals after consuming fava beans and is the leading cause of acute hemolytic anemia in affected patients [75,76].

Consumer acceptance of faba bean products remains low due to their distinct beany flavor, often perceived as an “off-flavor” in meat and dairy alternatives. This taste is attributed to volatile compounds and bitter-tasting polyphenols, which can negatively impact the sensory appeal of faba bean-based foods. As a result, developing processing techniques to reduce or mask these flavors is crucial for increasing consumer acceptance [65,70,77]. Faba bean is one of the most extensively cultivated legumes worldwide. In 2020, global faba bean production reached 5.67 million metric tons, a notable increase from 4.35 million metric tons in 1990. This surge in production reflects a growing interest in faba beans, driven by their potential to enhance food security and promote environmental sustainability [78]. Additionally, the global area under faba bean cultivation has almost halved from 5.4 million ha in 1961 to 2.6 million ha in 2019, but its productivity has increased from 0.9 t/ha to 2.1 t/ha in the corresponding years [79].

Faba leaves have also gained attention as an unconventional vegetable. They are commonly consumed in many countries as a vegetable similar to spinach, and they are used in several domestic cooking methods for consumption in salads, as a boiled vegetable, or baked in omelets. Faba leaves have also been used in traditional Chinese folk medicine for hemostasis and detoxication [80]. Symbiotic bacteria living in the root nodules of leguminous plants, such as *Vicia faba*, have the ability to fix atmospheric nitrogen (N_2_) and convert it into forms that are accessible to plants, significantly influencing their growth and development. They also contribute to lower CO_2_ emissions due to their ability to fix atmospheric nitrogen, reducing the need for nitrogen fertilizers [64,65,81].

#### 3.1.4. Lupin

Lupin is a genus of legumes belonging to the Fabaceae family, consisting of 200 species. The most commonly consumed *Lupinus* spp. are *L. albus* (white lupin), *L. angustifolius* (narrow-leafed lupin), and *L. luteus* (annual yellow lupin) [82,83]. Lupin seeds are rich in desirable nutrients such as protein and poor in digestible carbohydrates. Protein content, depending on the species and genus, ranges from 31 to 52% of dry matter. Proteins derived from lupin seeds are of increasing interest as a source of bioactive peptides. Lupin is also a very good source of dietary fiber, particularly its insoluble fraction. The content in dry matter amounts to approximately 40% [82,84,85]. Narrow-leafed lupin flour contains 7 to 12% of fat. The fatty acid composition of narrow-leafed lupin is characterized by high levels of oleic, linoleic, linolenic, palmitic and stearic acids [85]. Lupin seeds are also characterized by the high content of vitamins such as thiamine, niacin, riboflavin, tocopherols and other micronutrients, e.g., carotenoids, iron, zinc and manganese [86]. In human and animal studies, positive, health-promoting effects of this plant were observed, mainly for species *L. albus* and *L. angustifolius*. Those include antioxidative, anti-inflammatory, and hypotensive properties, lipid profile improvement and glucose metabolism regulation [82,87,88,89,90,91,92,93,94]. The health-promoting effects of lupin seeds are attributed to the chemical components that occur in them, e.g., phenolic compounds, phytosterols, squalene, polyphenols, carotenoids, bioactive peptides and dietary fiber [85,95,96]. Its versatile properties and its possible uses are becoming increasingly popular, which is why it should draw the attention of food producers to use it in the food industry for the production of functional foods. Studies suggest that the consumption of products enriched with lupin fiber has a positive effect on intestinal function and modification of risk factors for colon cancer [97]. Research showed a positive effect of products such as cookies, bread and pasta enriched with lupin protein on lipid profile and blood pressure in healthy individuals and those with hypertension [83,98,99,100]. Moreover, a diet enriched with lupin contributed to the improvement of glycemic profile in healthy individuals and diabetics [101,102,103,104]. Furthermore, researchers suggest that functional foods based on bioactive peptides of lupin grains have a pleiotropic effect on atherosclerosis, including oxidative stress, inflammation and cholesterol metabolism [105].

Non-nutritional compounds found in lupin seeds include lectins and protease inhibitors, polyphenolic compounds (condensed tannins, cyanogenic glycosides, and saponins), alkaloids, phytic acid, saponins and selected oligosaccharides. Antinutritional compounds are natural substances found in plants that can reduce the bioavailability of nutrients or have adverse effects on the organism. In excessive amounts, they may cause toxicity, impair the absorption of proteins, vitamins, and minerals, and trigger allergic reactions. However, these compounds can be deactivated or reduced to safe levels when proper preparation and processing techniques are used. Methods used to remove or reduce these non-nutritional compounds include cooking, sprouting, soaking, fermentation, selective extraction, enzymatic treatment, and irradiation [85]. Despite the antinutritional substances, the broad health-promoting properties and high nutritional value of lupin seeds make it a very good alternative to traditional sources of animal protein.

### 3.2. Hemp

*Cannabis sativa* L., known as hemp, is considered one of the oldest cultivated plants. It is an anemophilous herbaceous plant belonging to the hemp family. Hemp seeds are round, brown or dark red, about 3 to 5 mm. They can be consumed as whole, peeled seeds or in the form of processed products such as oil, flour, and protein isolate [106]. Although the hemp variety and environmental conditions have an impact on the composition and nutritional value of the seeds, they typically contain about 20–25% easily digestible and amino acid-rich protein and 25–35% fat, of which 90% are unsaturated fatty acids. Hemp oil is a rich source of essential unsaturated fatty acids, particularly linoleic acid, α-linolenic acid, and oleic acid. Carbohydrates constitute about 20–30% of the weight of hemp seeds, the majority of which is dietary fiber, both soluble and insoluble in a 20:80 ratio. Insoluble fiber of hemp seeds contains cellulose, lignin, and hemicellulose [107,108]. Moreover, hemp protein isolate is a source of all nine essential amino acids with a high content of glutamic acid and arginine and, at the same time, an adequate amount of sulfur-containing amino acids [107]. *Cannabis sativa* L. is a source of minerals such as potassium, phosphorus, magnesium, calcium, iron, zinc, and copper [106]. It is characterized by the high content of antioxidant compounds and other bioactive components such as phenolic compounds, bioactive peptides, carotenoids, tocopherols and phytosterols [108]. The highest content of phenolic compounds was observed in the shells, while the lowest was in the core. The dominant phenolic compounds in hemp seeds that exhibit antioxidant properties are lignanamides and hydroxycinnamic acid [107]. Hemp seeds contain all essential amino acids with particularly high levels of arginine, aspartic and glutamic acid, whereas lysine is considered as the limiting amino acid [109].

When blended with pea protein, it enhances the protein content and creates a more balanced amino acid profile, including increased levels of methionine and lysine. Meals prepared with these two ingredients provide a well-balanced amino acid profile [110].

Apart from a wide variety of essential nutrients, a number of antinutritional components can be found in this plant. Those include phytic acid, tannins, cyanogenic glycosides, trypsin inhibitors and saponins, which can negatively affect the nutritional value of the seeds [108]. In addition, hemp-based foods are required to meet standards for tetrahydrocannabinol (THC) content. The daily total THC intake, as reported by Fresh Hemp Foods Ltd., cannot exceed 1–7 μg/kg body weight, depending on the current requirements in a given country. It is important not to exceed the daily recommended dose due to serious side effects such as anxiety, paranoia, cognitive impairment, psychosis and addiction [111]. It has been observed that hemp protein digestion can contribute to the formation of bioactive peptides with antioxidative, antimicrobial, antihypertensive and cytomodulatory properties [112]. Moreover, the high level of polyunsaturated fatty acids in seeds, and especially in hemp oil, may contribute to reducing the risk of cardiovascular diseases, cancers, rheumatoid arthritis, hypertension, and inflammatory and autoimmune diseases [107]. Hemp seeds and their derivatives can be used as ingredients enriching food products, including bread, cookies, crackers, bars, meat and dairy products due to their nutritional and functional potential [108]. Adding hemp flour to wheat flour affects the properties of the dough and the quality of the bread. Bread enriched with hemp seed flour has an increased content of protein, fat, fiber, and microelements. However, larger amounts of hemp flour may worsen the volume and texture of bread, change its color to darker and increase the hardness of the crumb. Moreover, hemp seeds and their by-products can be used in the production of gluten-free, meat and dairy products, thus increasing their nutritional value and enriching their potential health benefits [107,108,111,113]. Hemp seed products are relatively new ingredients in the food industry, which poses a challenge for food manufacturers to produce a product with the desired sensory and technological characteristics while having high nutritional value at the same time. Although it can effectively improve the nutritional profile of some products or constitute a full-value product, further research is needed to determine the potential use of hemp seeds and their derivatives in the development of functional foods. It is particularly important to find a compromise between functionality and technological and sensory properties. To some extent, reducing the sensory value of enriched products in favor of health benefits may be acceptable to consumers looking for alternatives to traditional products.

Researchers have shown growing interest in hemp seeds due to their exceptional nutritional profile, potential health benefits, and widespread applications in human nutrition, including vegan and vegetarian diets. Additionally, their use as an ingredient in functional foods, dietary supplements, and pharmaceuticals further expands their role in health and medicine [108,114,115].

## 4. Aquatic Protein Sources

The fishing and aquaculture sector significantly contributes to global food security and nutrition. The global production was estimated at 178 million tons in 2020, with 63 percent coming from marine waters and 37 percent from inland waters. In 2020, 36 million tons of algae were produced, with 97% coming from marine aquaculture [116]. Seaweed and microalgae contain protein levels similar to traditional protein sources like meat, eggs, soy, and milk and have a higher protein yield per unit area [117]. Algae have a high growth and production rate, high photosynthesis efficiency, and low water usage, and do not compete for arable land. They also have the ability to store carbon dioxide. Moreover, oceans and seas cover more than 70% of the planet’s surface, allowing for sustainable cultivation and harvesting of large quantities of this potential resource [2].

### 4.1. Microalgae

Microalgae are some of the oldest organisms on Earth. Their role in the ecosystem is of great importance as they are responsible for the production of about 60 percent of the Earth’s oxygen [118]. The biomass of the cyanobacteria *Arthrospira* spp., known as “Spirulina”, and the green microalgae *Chlorella* spp. has been commercially produced on a large scale for food and feed applications in the early 1960s [119]. Microalgae have extraordinary nutritional quality, being a source of protein, peptides, carbohydrates, lipids, vitamins, carotenoids, long-chain omega-3 fatty acids, minerals, and other valuable trace elements. The percentage content in dry mass, depending on the species and cultivation parameters, is 6–71% protein, 8–64% carbohydrates, and 2–22% lipids [119,120,121].

Essential amino acids, including histidine, threonine, lysine, valine, isoleucine, leucine, phenylalanine, and methionine, are present in varying concentrations across different microalgae species [122]. An important role of microalgal proteins in human nutrition is their ability to complement the essential amino acid profiles of plant and pulse proteins, which often have different limiting amino acids. Methionine, cysteine and tryptophan are typically the limiting amino acids in most pulses, such as beans, peas, and lentils. In contrast, histidine or isoleucine are the first limiting amino acids in microalgae. Consuming these protein sources throughout the day improves the overall bio-efficiency of the amino acids in the pooled protein product [123].

There are two types of large-scale cultivation systems for microalgae production: closed photobioreactors and open ponds. Open ponds are shaped like oval racetracks. Their advantage is lower cost, but they have lower overall production rates and a higher risk of contamination. Closed-system photobioreactors are typically used for higher-value products or when biological protection is needed to ensure the system’s sterility or to prevent the release of foreign or genetically modified strains into the environment [124]. The nutrient profiles of commercial microalgae biomass vary greatly between and within species. This variability is likely due to different cultivation conditions used at various facilities. Microalgae products generally have a high protein content with a beneficial amino acid composition and a wide range of minerals [125]. Microalgae are rich in immunomodulatory compounds such as sulfated polysaccharides, sulfolipids, omega-3 fatty acids, and astaxanthin, which can activate the immune system to reduce inflammation in the body, combat bacterial infections, and inhibit the growth of cancer cells [126]. The study of Sandgruber et al. showed that *Chlorella pyrenoidosa* is a good source of vitamin D2 and may have a positive effect on blood cholesterol levels [121].

The vitamin B12 content in *Chlorella* products can vary significantly, ranging from <0.1 to 400 µg per 100 g of dry weight. However, it should be noted that some of the products contain inactive corrinoid compounds, such as 5-methoxybenzimidazolylcobamide and cobalt-free corrinoids, which do not exhibit vitamin B12 activity at all. This underscores the importance of accurate identification and analysis of these compounds, particularly when *Chlorella* is supposed to act as the sole source of vitamin B12 [127]. While microalgae like *Chlorella* and *Nannochloropsis gaditana* are potential sources of active vitamin B12, Spirulina predominantly contains pseudovitamin B12, which is not bioactive in humans [128]. Additionally, since production often uses surface or groundwater, microalgae can accumulate heavy metals, polycyclic aromatic hydrocarbons, toxins, pathogens, and pesticides [119]. While this ability can support environmental cleanup efforts, it also poses significant health risks, highlighting the need for effective purification processes to ensure safety [129].

Microalgae have not yet gained significant use as food or a substitute for conventional food. The main obstacles are the powdered consistency of dried biomass, its dark green color, and its slightly fishy smell, which limit its use in traditional dishes [130].

Studies indicate that *Chlorella vulgaris* biomass in small amounts improves dough rheological properties and enriches the product with protein, but in higher concentrations, it deteriorates texture sensory attributes and accelerates bread staling. In the case of muffins, consumer acceptance was high, while pretzels with 0.8% microalgae had the best sensory properties [131].

The volatile organic compound profile of microalgae and cyanobacteria, including sulfur compounds, branched hydrocarbons, and acyclic diketones, is highly strain-dependent and affects their aroma and taste. Understanding these substances is crucial for food applications, especially in utilizing marine-like flavors to create innovative products [132].

Microalgae are a promising raw material for feeding large populations, as they are used as innovative ingredients in the production of functional foods such as bread, pasta, or plant milk, which allows for the improvement of their technological properties and nutritional value. Additionally, microalgae offer numerous health benefits, such as antioxidative, cholesterol-lowering, anti-inflammatory, antihypertensive and anticancer properties, as well as supporting eye and cardiovascular health [133].

Microalgae farming represents a promising alternative to conventional protein sources, offering protein content up to twice as high as traditional resources [134]. It requires significantly smaller cultivation areas, and their production can utilize by-products from agriculture and industry. Additionally, microalgae are capable of absorbing 10 to 50 times more CO_2_ than land plants, making them an ideal solution [123] to meet the growing food demand while minimizing environmental impact [135,136].

Recent studies have highlighted microalgae as promising alternative protein sources due to their high-quality protein content and well-balanced amino acid profiles. These aquatic organisms offer a sustainable solution to meet the growing demand for protein-rich foods, especially as traditional animal-based protein sources face environmental and health challenges [123].

### 4.2. Macroalgae

Macroalgae include red, green, and brown algae. The most well-known are wakame, nori, and kombu, which are consumed fresh, dried, and pickled. Algae-derived agar-agar and carrageenan are often used as stabilizers, thickeners, and gelling agents in the food industry [118]. Seaweeds are known for their rich nutritional profile, which includes vitamins, minerals, antioxidants, and other beneficial compounds. Although the protein content can vary depending on the type of seaweed, it typically ranges from 10% to 30% of their dry mass. Seaweeds are considered a valuable source of plant protein, especially for those on a vegetarian or vegan diet, as they provide essential amino acids necessary for health [137]. In certain macroalgae species, sulfur-containing amino acids have been identified as the primary limiting amino acids. Similarly, tryptophan was found to be the first limiting amino acid across all species. However, macroalgae contain varying concentrations of essential amino acids, including histidine, threonine, lysine, valine, isoleucine, leucine, phenylalanine, and methionine. In particular, red species like *Porphyra* and *Gracilaria* exhibited high levels of free alanine, glutamic acid, and aspartic acid, contributing to their unique nutritional profile [138,139].

Marine algae (seaweeds) live in the intertidal zone on the bottom of water bodies. They are characterized by autotrophic nutrition and rapid growth; they do not need soil for cultivation, and their growth rate is faster than that of land plants. They are divided into three different groups: Chlorophyta (green), Phaeophyta (brown), and Rhodophyta (red), based on their pigment composition and stored nutrients [140]. Most seaweeds intended for industrial use are cultivated, with only a small amount coming from wild harvests [141]. In their natural state, algae contain 80–90% water. On a dry weight basis, they consist of approximately 50% carbohydrates, 1–3% lipids, and 7–38% minerals. The protein content is highly variable (10–47%) and includes a high amount of essential amino acids. Algae contain more vitamins A, B12, and C, β-carotene, pantothenate, folic acid, riboflavin, and niacin than common terrestrial fruits and vegetables [142,143]. A systematic review and meta-analysis by Vaughan et al. (2022) [144] demonstrated that brown seaweeds and their extracts have the potential for T2D prevention and treatment, both through consumption and supplementation. Additionally, the meta-analysis confirms that brown seaweeds positively affect plasma glucose homeostasis [144]. Similar conclusions were demonstrated in the meta-analysis published by Kim et al. (2023) [145]. The results showed that incorporating seaweeds into the diet can potentially help in the prevention and T2D management by improving blood sugar control. The meta-analysis of randomized controlled trials found that the group consuming seaweeds had reduced postprandial glucose levels, HbA1c, and HOMA-IR compared to the control group [145].

Although seaweed consumption offers numerous health benefits, it also carries the risk of accumulating harmful substances, such as heavy metals and pathogenic microorganisms. This may negatively impact human health and have various consequences [80]. Consuming up to 5 g of dehydrated seaweed daily is considered safe for adults. Further, in vivo research is essential to evaluate the bioavailability of trace elements and toxic metals, determine the actual concentrations absorbed during seaweed ingestion, and assess the potential health risks associated with excessive consumption [146]. Europe has recently been recognized as one of the most innovative regions for using seaweed as a food ingredient. According to the Seafood Source report, new products containing seaweed introduced to the European market increased by 147% between 2011 and 2015, making Europe the most innovative region in the world after Asia [147].

Recent studies have highlighted the potential of macroalgae as sustainable and nutritious protein sources, offering a viable alternative to traditional animal-based proteins. To fully harness their potential, further research is needed to optimize protein extraction methods and address challenges related to large-scale production [148,149].

### 4.3. Water Lentils (Duckweed)

Duckweed is the smallest and fastest-growing flowering plant in the world, capable of producing enormous biomass with a wide range of potential applications, such as food production, biofuels, and biogas [150]. Duckweed exhibits high physiological efficiency and longevity. The family Lemnaceae comprises 37 species classified into five genera, divided into the subgroups Lemnoideae (Spirodela, Landoltia, and Lemna) and Wolffioideae (Wolffiella and Wolffia). These plants are found worldwide and are adapted to diverse ecological conditions, tolerating a wide pH range (from 3 to 10.5) and temperatures (from 17 to 30 °C). Duckweeds can double their biomass within 96 h, making them an excellent resource for various applications, including as a food source [151]. The cultivation of water lentils used for the production of novel foods is conducted in greenhouse ponds without the use of pesticides [152]. Duckweed, particularly Wo. globosa has traditionally been used as a food source for humans in some Asian countries, such as Thailand, Laos, and Cambodia. Certain species have an exceptionally high protein content, ranging from 35 to 40% of dry mass, and contain a spectrum of essential amino acids comparable to soy. The essential amino acid profile of duckweed is more balanced than most plant proteins and closely resembles that of animal protein. Its nutritional value is comparable to alfalfa, particularly in terms of lysine and arginine. Additionally, duckweed contains high levels of leucine, threonine, valine, isoleucine, and phenylalanine. However, methionine and tryptophan were identified as the primary limiting amino acids across all species. It is recommended to combine with seeds such as hemp or with cereals to complement the amino acid profile [151,152].

The levels of starch, protein, fat, minerals, vitamins [153,154], phytosterols, amino acids, and fatty acids in duckweed align with the dietary recommendations of the World Health Organization (WHO) for human nutrition [155]. The study by Pagliuso et al. (2022) [151] analyzed 21 ecotypes of 14 species from the subfamilies Lemnoideae and Wolffioideae for their nutritional components, such as starch, soluble sugars, cell wall components, amino acids, phenols, and tannins. These findings, combined with data from 85 other publications, suggest that duckweed have nutritional profiles comparable to common food sources and can be used for human consumption worldwide beyond their current use in Asia [151]. European Food Safety Authority (EFSA) Panel on Nutrition, Novel Foods and Food Allergens (NDA) has issued an opinion on the protein concentrate from water lentils (mixture of *Lemna gibba* and *Lemna minor*) that resulted in registration as a novel food in accordance with European Union (EU) Regulation 2015/2283. The Panel stated that the water lentil protein concentrate from the mixture is safe as a food supplement for adults with a low risk of causing allergic reactions [152]. However, some species of duckweed, such as those from the genus Lemna, contain significant amounts of calcium oxalate in the form of raphides, which can be associated with health issues like kidney stones. Those compounds are not present in rootless duckweeds such as *Wolffiella* spp. In addition to their rapid growth rate, duckweed exhibits remarkable characteristics such as a natural aquatic habitat (eliminating the need for arable land), small size, and floating lifestyle, which facilitate easy harvesting. These features make water lentils a sustainable and ecological alternative to traditional farming systems [155].

Research into duckweed as an alternative protein source has gained increasing attention due to its rapid growth, high protein content, and environmental sustainability [156]. However, further studies are needed to optimize its production, improve its nutritional profile, and address challenges related to large-scale commercialization. Duckweed shows potential as a valuable addition to the global protein supply, especially in the context of rising demand for plant-based protein alternatives.

## 5. Other Alternative Protein Sources

### 5.1. Insects

Entomophagy, meaning the consumption of insects by humans, is a more environmentally friendly approach to increasing protein intake from non-traditional sources while contributing to global food security [157]. In many cultures, insects have been considered not only as a source of food but also as a delicacy since ancient times. They are consumed raw or processed, either whole or as an ingredient in food products [158]. In accordance with EU Regulation 2015/2283 on novel foods, the mealworm (*Alphitobius diaperionus*), the house cricket (*Acheta domesticus*), the yellow mealworm (*Tenebrio molitor* larva) and the migratory locust (*Locusta migratoria*) are officially approved for commercialization in the European Union [159]. For food production purposes, insects are raised in controlled conditions (temperature, humidity and light) in special containers such as cages, boxes or terrariums. Organic waste or specialist feeds are used to feed the insects, supporting a circular economy and reducing waste. Once they reach the appropriate size, the insects are collected, cleaned and processed. Processing can involve cooking, drying or grinding into a protein meal. In some cases, the wings and legs are removed to improve the quality of the final product. Insects are processed into various forms, such as flours, protein powders, bars or other food products. To ensure food safety, these products are sterilized [160].

Studies have shown that the nutritional quality of edible insect species is very high, as their protein content ranges from 67 to 72 g per 100 g dry weight, with a complete profile of essential amino acids, particularly high in leucine. The limiting amino acids are usually methionine, tryptophan and lysine [159]. It is recommended that this protein source be combined with seeds such as hemp, legumes, and cereals to complement the amino acid profile [161].

Insects are also a source of fat, fiber and minerals [162]. The total protein digestibility is comparable to that of traditionally consumed meat, estimated at 77–98. Edible insects are reported to contain fewer saturated fatty acids and more monounsaturated and polyunsaturated fats compared to meat. Their total fat content ranges from 1 to 57 g per 100 g. The presence of chitin, which serves as a source of dietary fiber, may improve the lipid profile; however, higher levels of this polysaccharide are correlated with reduced protein digestibility [162,163]. An in vitro study demonstrated that iron from insects such as grasshoppers, crickets, and mealworms exhibits higher chemical solubility compared to iron from beef, indicating their potential as an excellent source of bioavailable iron [164]. Incorporating edible insects into the diet can enhance dietary diversity and support proper nutrition for individuals with high nutritional needs, such as teenage girls, pregnant women, the elderly, and the malnourished. Adding 100 g of dried crickets, mealworms, or locusts provides the recommended daily intake of zinc, copper, and phosphorus. Replacing 10% of wheat flour with cricket powder in bread and pasta recipes increases zinc content by approximately 90–100%. Edible insects are also a source of vitamins B2, B5, B7, and folic acid [165].

Individuals unfamiliar with the concept of eating insects may struggle to accept them, especially in their whole form. Performed research suggests that proper labeling highlighting the benefits of insect consumption, along with an aesthetically appealing presentation, can increase the acceptance of insect-based products [166]. Studies have shown that insect-based foods can have a beneficial mineral content with low levels of contamination by metals, metalloids, and rare earth elements (REE), posing no safety concerns. Only a few samples in feed products have exhibited higher levels of toxic elements, such as aluminum (Al) and lead (Pb), which should be carefully monitored during production [167]. Insect production places significantly less strain on the environment compared to traditional protein sources and requires much less water than conventional livestock farming.

Consumption of edible insects is particularly common in African countries. Studies have shown that it could serve as a path to improving environmental health and food systems by reducing meat consumption. In the long term, it will also contribute to lowering greenhouse gas emissions, such as CO_2_ and methane, associated with livestock production. Edible insects are rich in proteins, fats, and iron, which can help combat malnutrition and food insecurity. Moreover, their production has a lower carbon footprint and requires fewer natural resources, such as land and water, compared to livestock farming [168]. The use of insects for waste recycling is particularly timely, as it is estimated that around two billion tons of solid waste (at least 50% of which is food and green waste) are generated globally each year, with only 33% being efficiently recycled. The need for food production technologies that require limited resources (land, water, and labor) and have a minimal environmental impact highlights the growing importance of edible insects in the circular economy and regenerative agriculture to ensure the sustainability of global food systems [169]. For example, crickets need only about 1 L of water to produce 1 kg of body mass, whereas cattle require up to 15,000 L per kilogram of meat [170]. Insect farming generates significantly fewer greenhouse gases compared to beef, pork, or poultry production. Crickets, for instance, emit up to 80% less methane and CO_2_ than cattle [171]. Insects are more efficient at converting feed into protein—crickets need only 1.7 kg of feed to gain 1 kg of body weight, while cattle require 8 kg. This makes their production more sustainable [172]. Scientists have proposed using insects to reduce food waste. Industrially farmed insects can effectively transform several tons of food waste into valuable products, including human food, animal feed, fertilizer, and secondary industrial compounds (e.g., biofuels, lubricants, pharmaceuticals, and dyes) [173].

Research on edible insects has gained momentum as a potential alternative protein source for both human consumption and animal feed. Insects are being explored for their high nutritional value, including protein and essential nutrients while offering a sustainable and eco-friendly solution to feed a growing global population. However, challenges related to mass production, safety, and affordability need to be addressed to develop a reliable insect farming industry [174,175].

### 5.2. Microbial Proteins

Microorganisms have always played a key role in fundamental food processing techniques, such as dough fermentation for bread production or the processing of milk into cheese, enabling its long-term storage. Microbial protein represents an innovative, alternative source of high-quality protein that can replace animal-derived proteins. It meets FAO/WHO standards for essential amino acids in human nutrition, making it a valuable dietary component for direct human consumption [176]. Protein can be obtained by cultivating microbes that contain over 30% of the protein in their biomass and offer a balanced profile of essential amino acids. This microbial protein is commonly known as single-cell protein (SCP), although some producers, like filamentous fungi and algae, are multicellular. Besides direct consumption as SCP, microbes also enhance protein content and quality in fermented foods [177]. SCP is obtained from the cells of certain strains of microorganisms, such as yeast, fungi, algae, and bacteria, which are cultivated on various carbon sources for synthesis [178]. Microbial foods are produced in three ways. The first involves the production of microbial biomass from production waste or gas in a bioreactor. The second method is the fermentation of substrates inoculated with microorganisms. The third method relies on genetic engineering, where the final product is a purified substance [179]. SCP is produced through a sequential process that begins with substrate preparation, where raw materials like fruit, vegetable waste, or lignocellulosic materials are converted into a fermentation-ready medium. Depending on the substrate type, methods like wet, dry, or direct preparation are used, followed by sterilization to create a nutrient-rich medium. The prepared medium is then inoculated with microorganisms, which ferment the substrate, efficiently converting it into microbial biomass rich in protein. After fermentation, the biomass is harvested, purified, and processed to yield SCP as a valuable protein source [178].

Microbial proteins have high nutritional value because they contain all eight essential dietary amino acids [180]. The protein content in dry matter varies across sources: fungi contain 30–45%, microalgae 40–60%, yeast 45–55%, and bacteria 50–65% [178]. Besides proteins, SCP includes carbohydrates, nucleic acids, fats, minerals, and vitamins. It is also a good source of B vitamins and polyunsaturated fatty acids, such as linoleic acid [181,182]. When it comes to fat content, bacteria have the lowest levels at 1–3%, while microalgae boast the highest, ranging from 7 to 20% of dry matter [178]. They often contain higher levels of essential amino acids, such as lysine and methionine, which are typically limited in plant proteins [176,180,183].

Although microbial proteins are rich in nutrients, they are not pure proteins but contain other microbial cell components, which may cause food safety concerns. Some yeast and bacterial proteins have a high nucleic acid content, especially RNA. High amounts of those compounds can lead to the formation of uric acid, which causes gout. Research is needed to develop technologies for nucleic acid removal [180,184]. SCP microorganisms have immense potential to provide nutrition for larger populations. Advances in cultivation, including the fed-batch process, enabled the production of microbial biomass from affordable waste streams into protein-rich biomass at the beginning of the 20th century [179]. Microbial proteins can be produced year-round, independent of climate or season, and do not require large areas for cultivation. This rapid production capability is advantageous for meeting urgent food demands [185].

A new type of alternative proteins has emerged on the U.S. market—animal proteins produced recombinantly through microbial fermentation. Precision fermentation is an advanced biotechnological technique in which genetically modified microorganisms such as bacteria, yeast or fungi are used to produce specific biomolecules such as proteins, fats, enzymes or dyes. That method enables the production of animal-derived proteins, such as milk and egg proteins, using only microorganisms. This innovative method offers a sustainable alternative to traditional animal agriculture, significantly reducing the ecological footprint associated with protein production [186,187]. Currently, EFSA has not yet approved any products derived from precision fermentation for introduction to the EU market [188]. The cultivation of microorganisms stands out as an environmentally sustainable process due to its minimal land requirements and efficient resource utilization. Waste materials, such as spent yeast or algae from biorefineries and the brewing industry, are often repurposed as substrates for production, reducing environmental waste streams [1]. Additionally, methane—a by-product of animal farming and biogas production—is gaining traction as a valuable substrate for producing SCP. Companies leverage methanotrophic bacteria and advanced fermentation technologies to transform methane into high-protein feed products. This innovative approach not only mitigates methane emissions, a potent greenhouse gas but also promotes circular economy principles, underscoring the environmental benefits of SCP production [176,189]. Mycoproteins are currently the most popular single-cell protein used to produce a variety of foods. For this reason, we decided to describe them in more detail below.

#### Mycoproteins

Mycoproteins, also known as mycelium-based proteins or fungal proteins, are a form of single-cell protein derived from fungi for human consumption. They are gaining increasing attention as an alternative to animal-derived protein. This group of proteins primarily includes proteins derived from fermented biomass produced by filamentous, eukaryotic, soil-dwelling, non-pathogenic fungal microorganisms. *Fusarium venenatum* is one of the main strains used in the cultivation and harvesting of mycoproteins [190]. This protein was commercially developed in the 1980s and has been recognized as safe for consumption by the FDA [191]. Quorn^®^ is a well-known mycoprotein product available in many countries worldwide, produced through the fermentation of fungal spores with glucose and other nutrients [192]. Mycoprotein has a biological value comparable to that of animal proteins, with similar texture, sensory characteristics, and high digestibility. On average, 100 g of dry mycoprotein contains 45% protein, 13% fat, 10% carbohydrates, and approximately 25% dietary fiber. Fungal protein has an amino acid composition comparable to that of milk and egg, with a particularly high lysine content [193]. The fiber in fungi primarily consists of β-glucan (up to 75%) and chitin. The European Commission classifies myco-protein as “food high in dietary fiber” [194]. Furthermore, mycoprotein contains vitamin B12, folic acid, zinc, magnesium, calcium, and phosphorus. The lipids present in mycoproteins mainly include linoleic and linolenic acids, with minimal levels of saturated fatty acids and cholesterol [190,195].

Numerous nutritional and health benefits of mycoproteins have been demonstrated in studies. They contribute to improved appetite regulation through effects on metabolic and satiety hormones and present positive effects on lipid profile and blood pressure. Additionally, they reduce the risk of cardiovascular diseases and exhibit antihyperlipidemic, antioxidative, and antimicrobial properties while stimulating muscle protein synthesis [190,192,194,195,196]. However, despite their classification as safe for consumption, mycoproteins may potentially cause allergic and gastrointestinal symptoms such as anaphylaxis, hives, nausea, vomiting, and diarrhea. Because they share allergenic traits with *Aspergillus fumigatus* and *Cladosporium herbarum* (and to some extent with *Alternaria alternata*), individuals allergic to molds may also experience allergic reactions after their consumption. Mycotoxins in fungi can also be a concern, but the strains used in mycoprotein production typically do not produce them, and this is assessed during safety testing [192,195].

Fungal proteins exhibit excellent emulsifying, gelling, and foaming properties, making them widely used in the food industry. They are primarily utilized in the production of meat substitutes such as minced meat, chicken chunks, sausages, burgers, nuggets, pies, pastries, and ready-made meals, as in the form of frozen foods [195,197,198,199,200]. Research assessing the environmental impact of mycoprotein production is still limited, and its environmental implications remain controversial. Some studies indicate that mycoproteins represent a low-impact alternative protein source in terms of water, carbon footprint, and land use [185,201]. However, many recent studies challenge these claims, applying more detailed analyses and comparing mycoproteins with other protein sources. One study found that the global warming potential of mycoproteins is comparable to that of egg proteins but significantly higher than soybean meal. The authors also noted that further processing of fungal biomass could double the environmental impact of mycoproteins [103]. Another review showed that mycoproteins were rated worse than poultry and fishmeal concerning potential global warming effects [202]. Life Cycle Analysis (LCA) has revealed a high environmental impact of myco-protein products, primarily due to the energy-intensive cultivation process [203]. Another LCA study showed that mycoproteins have a similar environmental impact to pork and chicken [204].

The nutritional properties of mycoproteins and their applications in the food industry provide a solid foundation for their use as an alternative to animal-derived protein sources. However, to fully harness their potential and meet the demands of increasingly health-conscious consumers, further interdisciplinary research will be necessary.

### 5.3. Cultured Meat

The ability to take a small number of cells from living animals and grow them in a controlled environment to produce food is an emerging field in food science. Advances in cell culture technology allow cells from farm animals, poultry, seafood or other animals to be used in food production [205]. The U.S. Food and Drug Administration (FDA) announced its first approval of a lab-grown meat product on 16 November 2022. The product, UPSIDE Foods’ lab-grown chicken, was deemed safe for human consumption [206]. The process of producing cell-cultured meat and other agricultural products using cell culture technology is called cellular agriculture. In the case of cultured meat, stem cells are grown in a medium containing all the required nutrients, such as hormones, growth factors, and fetal bovine serum, to enable them to divide and proliferate [207]. Skeletal muscles produced in a bioreactor are composed of muscle cells connected by connective tissue [208]. Other types of cells, such as adipocytes, fibroblasts, chondrocytes, and endothelial cells, are also cultured in parallel to enhance the sensory and structural properties of the meat [209]. The process of producing cultured meat begins with collecting a sample of cells from tissue which does not harm or kill the animal. Some cells from the sample are cultivated to create a “cell bank” for future use. A small number of cells are placed in a tightly controlled environment (usually in several closed, sterile vessels of increasing size) that supports cell growth and proliferation by providing the appropriate nutrients and other factors. Once the cells have multiplied into billions, additional substances (such as protein growth factors, new surfaces for cell attachment, or extra nutrients) are introduced into the controlled environment to enable the cells to differentiate into various types and develop characteristics of muscle, fat, or connective tissue cells. When the cells have differentiated into the desired type, the cellular material can be harvested and prepared using conventional food processing and packaging methods [205].

Cultured meat aspires to be biologically equivalent to traditional meat. It contains easily digestible proteins, excellent amino acid composition, vitamins and minerals [210].

The amino acid composition of cultured meat depends on the amino acid profile of the cell used for proliferation. However, differences exist in the levels of total and free amino acids, as well as nucleotide-associated compounds, compared to conventional meat. Nevertheless, its overall amino acid composition in relation to protein content is comparable, if not superior. Additionally, the supplementation of amino acids in the culture medium facilitates their absorption during the formation of cultured meat [211,212].

The lack of a circulatory system can lead to nutritional limitations in the product. To improve its nutritional quality, research is being conducted on incorporating collagen, which is rich in essential amino acids, as well as scaffolds made from algae, plants, and fungi as sources of dietary fiber and additional health benefits. At the same time, the introduction of adipocytes allows for an increased fat content in the meat. Studies are also underway to address deficiencies in certain fatty acids, such as alpha-linolenic and linoleic acids, as well as amino acids. Additionally, efforts are being made to enhance the content of myoglobin, which is responsible for the color and flavor of the meat, along with supplementary vitamins such as taurine. Further research is ongoing to optimize the full nutritional profile of cultured meat to closely match the quality of conventional meat [213].

Unlike animals raised for meat, cultured muscle cells are not exposed to intestinal pathogens, reducing the risk of foodborne illnesses. Disease outbreaks caused by *E. coli*, *Salmonella*, or *Campylobacter* affect up to a million people each year [214]. Controlled conditions and strict oversight also eliminate the need for large quantities of antibiotics, thereby reducing the risk of antibiotic resistance development [214,215]. A major challenge in the production technology of cultured meat lies in replicating its nutritional, structural, and flavor properties. The complex biochemical processes that occur after the slaughter of an animal lead to numerous changes in the meat structure. These processes give meat its characteristic flavor, texture, water-binding capacity, and taste properties. The processes occurring in cultured meat are not yet fully understood, making it difficult to assess the differences and similarities compared to conventional meat. Further research is required in this area, but cultured meat samples are currently difficult to obtain for scientific study [210]. A small-scale study showed that cultured chicken meat, compared to conventionally produced chicken meat, may have lower protein content, an overall poorer amino acid profile, less magnesium and vitamin B3, as well as higher levels of saturated fats, cholesterol, and heavy metals such as cadmium (Cd) and lead (Pb). Despite these differences, all levels were compliant with EU regulations, and the meat was deemed safe for consumption [216]. Promising results have been observed in studies conducted in several European countries, where over 50% of respondents expressed a positive attitude towards cultured meat, emphasizing its reduced cruelty to animals and their willingness to purchase it [217,218,219]. Currently, in vitro meat is not available in Europe.

Production of cultured meat takes much less time than in the case of traditional breeding—the tissue can be collected within a week instead of months (as in chickens) or years (as in pigs and cows). Thanks to this, the energy and work requirements per kilogram of in vitro meat are significantly lower [213]. Cell-based meat offers a promising, environmentally sustainable source of high-quality protein. It requires approximately two-thirds less land, allowing the reclaimed space to be repurposed for environmental restoration and renewal [215,220,221]. Emissions related to nitrogen and air pollution are also lower due to higher efficiency and production in a closed system without manure. Using renewable energy, the carbon footprint is lower than in traditional meat production. However, the energy used to maintain the temperature in the reactors and for the biotechnological production of the culture medium ingredients is high, and these parameters need to be improved to reduce the impact on the climate [215,221].

Advancements in cultured meat research highlight its potential to address global food security and sustainability challenges. A bibliometric analysis of 484 articles from 2000 to 2022 reveals significant progress in cell-cultured meat research, with notable contributions from various countries and institutions [222]. To fully realize the potential of cultured meat, future research must focus on optimizing production efficiency, reducing costs, and ensuring consumer acceptance. Such efforts are crucial to establishing cultured meat as a viable and sustainable alternative protein source [223,224].

## 6. Alternative Protein Sources—Ethical, Environmental and Health-Related Aspects

### 6.1. Ethical Aspects Related to Protein Consumption

The consumption of animal-derived protein has expanded to an unprecedented scale. Everyday, approximately 900,000 cows, 202 million chickens (140,000 per minute), and 3.8 million pigs are slaughtered for meat. The number of fish killed is so vast that it is challenging to estimate, amounting to hundreds of millions each day [225]. Additional concerns arise from the conditions in which these animals are raised. Overcrowding on small surfaces prevents basic movements, such as chickens being unable to stretch their wings [226]. In recent decades, public concerns regarding livestock farming have increased significantly in Western countries. Animal well-being seems to be an important factor to contemporary consumers as studies show that they prioritize it over sustainability and environmental aspects. According to customers, animal-friendly farming is expected to have a positive impact not only on the animal but also on human health and is associated with higher-quality products [227]. Meta-analysis of Consumer Studies carried out by Janssen et al. in 2016 showed that consumers believed that outdoor access, stocking density and floor type have an impact on animal welfare [228]. On average, they had a positive attitude toward more animal-friendly husbandry systems with outdoor access and were willing to pay more for products based on those systems [228]. At the same time, ProVeg’s 2021 survey report showed that European consumers are moving towards plant-based foods due to health reasons, animal welfare and environmental concerns [229]. Whereas most alternative protein sources do not raise any ethical concerns, products such as cultured meat or insects may be open to doubt. Current methods of manufacturing cultured meat involve biopsies for stem cells, and its production is associated with global food corporations, which may impact local self-sufficiency. Moreover, cultured meat may be perceived as something unnatural and might potentially open the door to cannibalism [230]. There is a limiting amount of data regarding consumer acceptance related to insect consumption [231]. According to Fukuda et al., 74% of studied US respondents (*n* = 361) thought insects could feel pain [232]. Insect welfare regulations, which consider different aspects, including housing, feed, transport and killing, are needed to address ethical issues associated with insect farming.

### 6.2. Impact of Meat Protein and Their Alternatives on Environment

Meat production significantly contributes to antibiotic resistance, extensive land use, and greenhouse gas emissions [220]. It is estimated that in 2020, 30% of global greenhouse gas production was associated with food systems. Fifty-seven percent of those emissions were derived only from the production of red meat and milk. Large emissions of greenhouse gases lead to negative climate changes that are responsible for damaging natural and human systems. Increasing average world temperature seriously threatens water security, which is critical for human and environmental health [233]. Moreover, animal-based food production is associated with great water and land use. The increasing need for arable land results in deforestation, which has a deleterious impact on Earth’s health [234,235]. This phenomenon impacts not only climate but also biodiversity, as thousands of species may be in danger of extinction [236]. Another environmental aspect associated with meat production concerns the pollution of water, soil and air. Runoff from animal farms, use of fertilizers, methane production and soil erosion are among the factors responsible for the contamination of crucial environmental elements [237].

Alternative protein sources use a fraction of the land and water that is required for animal-based food production and generate significantly smaller amounts of greenhouse gases [238]. A study by Costa et al. showed that substituting beef meatballs with pea protein balls can save 2.42 kg of CO_2_ equivalent per serving (100 g) [239]. Moreover, research by Scarborough et al. showed that the dietary impact of vegans and vegetarians is significantly lower than that of meat-eaters in terms of greenhouse gas emissions, land and water use, eutrophication and biodiversity. The authors concluded that limited consumption of animal-based foods can make a substantial contribution to the reduction of the environmental footprint [240]. The current literature suggests that plant-based meat substitutes, on average, have a 50% lower environmental impact [241]. These findings highlight the potential of plant-based diets to mitigate environmental impacts and support the transition to more sustainable food systems.

### 6.3. Alternative Protein Sources—Health Benefits and Concerns

The 2024 ProVeg International report indicates that alternative meat products made from nutrient-rich ingredients are healthy, have a more favorable nutritional profile than traditional meat products, and contain more fiber and less saturated fat [242]. The negative impact of red and processed meat consumption is well-established. Research on big datasets showed that high intake of red and processed meat in Western countries is associated with higher mortality rates [243]. In 2015, the International Agency for Research on Cancer (IARC) stated that consumption of processed meat can be considered carcinogenic to humans. Red meat consumption was described as a probable carcinogen [244]. World Cancer Research Fund recommends limiting consumption of red and processed meat (350–500 g of cooked red meat and as little processed meat as possible) to prevent colorectal cancer [245]. Moreover, there is strong evidence linking red and processed meat consumption with cardiovascular disease and T2D [246].

Concurrently, plant-based diets have numerous health benefits. Vegetarians and vegans are usually characterized by lower body mass, blood pressure, LCL-C and HbA1c concentration. This is associated with an improved cardiometabolic profile and lower risk of T2D and CVD morbidity and mortality [247]. In the EPIC-Oxford study, non-meat eaters had a relatively lower risk of various chronic diseases, including ischemic heart disease, diabetes, diverticular disease and cataracts. At the same time, a potentially negative effect on health was observed due to insufficient intakes of vitamin B12, vitamin D and calcium [248]. The risk of nutritional deficiencies in plant-based diets should not be overlooked. Balancing a meat-free diet with appropriate nutritional intake is possible. However, it may require professional advice and, in some cases, the use of supplements (e.g., vitamin B12 for vegan diets). It also should be noted that the current market of plant-based products is rapidly evolving, with new categories of unhealthy meatless products appearing on market shelves—ultra-processed plant-based foods. All health benefits associated with vegetarian and vegan diets are linked to a high intake of whole grains, fruits, and vegetables and, in general, a low intake of ultra-processed foods. However, modern plant-based diets may not fit into these characteristics. Currently, consumers have a wide range of products that are supposed to be equivalent to goods perceived as tasty—meatless nuggets, burgers, sausages, plant-based “milk”, “cheeses” and desserts. Despite the lack of animal-based ingredients, they still have substances that negatively impact health, such as sugar, saturated fats, and food additives. Therefore, there is a need to classify plant-based diets as “healthy” and “unhealthy”, based on the intake of different food groups and products. This phenomenon has been recognized by other researchers [249], as well as WHO [247]. Consumers and health professionals should be made aware that intake of alternative proteins has many health benefits, but they will also depend on the types of products consumed.

A fairly new alternative to a completely meatless diet that emerged in recent years is the planetary health diet (PHD). This nutritional model was introduced in 2019 by EAT in a report entitled “Food in the Anthropocene: The EAT–Lancet Commission on Healthy Diets from Sustainable Food Systems”. PHD is based on unprocessed, local and mostly plant-based products and involves a low intake of saturated fats and an adequate intake of fiber. The suggested amount of meat intake is very limited, especially for red meat—a few grams a day. Appropriate protein intake is supposed to be achieved by the intake of alternative protein sources, including legumes, peanuts and tree nuts. Following those recommendations is supposed to have a positive impact on both human and environmental health [250]. Further studies confirmed those associations as follows: PHD has been associated with a lower risk of T2D, obesity, CVD and mortality [251]. The inclusion of alternative protein sources allows us to follow PHD recommendations. The intake of products described in this article supports the limitation of meat consumption while achieving appropriate protein intake and supporting environmental health.

Recent research highlights the need to reassess dietary recommendations considering nutrient bioavailability. One of the studies published in 2023 in the Lancet Planetary Health study found that, without supplementation, achieving micronutrient adequacy in a flexitarian diet requires doubling animal-source foods to at least 27% of total calories while reducing plant-source foods rich in antinutrients [252]. Another study suggests that 35% of calories from animal-source foods may be necessary for a nutritionally adequate diet [253]. Given the heterogeneity of results and methodological flaws in studies of the health effects of plant-based diets, rigorous, randomized, controlled trials of all newly proposed environmentally protective diets are needed. These studies should include validated biomarkers of nutritional status and assess the levels of supplementation and/or fortification that would be required to ensure adequate micronutrient and protein intake [254,255].

Replacing red and processed meat with plant-based proteins offers significant cardiovascular benefits, with a higher plant-to-animal protein ratio linked to lower CVD and CAD risk. While guidelines promote plant proteins for health and sustainability, most protein in developed countries still comes from animal sources. Further research is needed to determine the optimal plant-to-animal protein ratio for cardiovascular health [256].

In summary, consuming plant-based protein sources is associated with better health outcomes, particularly for the cardiovascular system, compared to animal-based products. The mechanisms behind these effects are not yet fully understood, highlighting the need for further research, including studies on plant protein digestibility. The undeniable health benefits of plant proteins align with their lower environmental impact, which should be considered when designing optimal diets. Human and planetary health are inseparably linked [257].

Health professionals should take on responsibility for integrating health considerations with environmental protection and climate change mitigation, as global food production—especially meat—significantly impacts these issues. At the same time, climate change affects human health, ecosystems, and food quality, creating a closed cycle of interdependencies. Therefore, future research should not only clarify the health effects of plant proteins but also promote better agricultural practices and influence public health policies in ways that benefit both people and the planet.

## 7. Conclusions

This article gathers current knowledge on available alternative protein sources and their impact on human and environmental health. Table 1 presents the overview of described products and their characteristics, including their nutritional value, environmental and health benefits, as well as possible limitations.

Technological and industrial development certainly contributed to the larger availability of alternative protein sources. Currently, customers can choose from a wide range of products—from traditional foods derived from plants, such as well-known legumes, to less obvious choices, such as aquafeed, insects, cell-cultured meats or microbial proteins. Available research demonstrates that choosing alternative protein sources has numerous advantages, including ethical, environmental and health benefits. Those products are characterized by beneficial nutritional profiles, which can positively impact human health and reduce the risk of various chronic conditions. Moreover, lower environmental impact, higher ethical standards and more efficient production can address the challenges of the modern world.

Organizations responsible for public health, food production and safety should focus on promoting sustainable food products, including alternative protein sources. At the same time, their limitations should be taken into account, including lower bioavailability of certain nutrients, the presence of antinutritional compounds, or problems related to consumer acceptance. Future technological advancements and optimization of production processes may address some of those limitations. Moreover, further scientific research is still needed to understand the nutritional value, safety, and long-term impact on human health of novel substitutes for animal-based foods.

## Figures and Tables

**Table 1 nutrients-17-01148-t001:** Overview of various protein sources and their characteristics, i.e., protein content (% dry weight), manufacturing methods, lipid content (% dry weight), nutritional value, benefits, and potential limitations.

References	Sources	Kg CO_2_e per kg	Protein % d.w.	Manufacturing Method	Lipid % d.w	Nutritional Value	Health Benefits	Environmental Benefits	Limitations
[32,38,44,45,46,47,48,258,259,260]	Soybean	0.38–0.85	35–40	Traditional cultivation	20	high-quality vegetable protein, soluble fibers, oligosaccharides, minerals, vitamin B, soy lecithin, bioactive phytoestrogens, low in saturated fat and a source of unsaturated and omega-3 (*n*-3) fatty acids	consumption is associated with lower levels of total cholesterol, low-density lipoproteins, and triglycerides; prevention and control of T2D; antioxidant properties; helps reduce insulin resistance; anti-inflammatory effects; supports bone health; reduced risk of cancer; helps alleviate menopause symptoms	Compared to animal protein sources, soybeans are more sustainable, requiring less land and water while producing lower greenhouse gas emissions, fixing atmospheric nitrogen, and reducing the need for synthetic fertilizers;	Negative impact on thyroid function in individuals with iodine deficiency or subclinical hypothyroidism.
[55,56,57,59,60,62,261]	Pea	0.8	20–25	Traditional cultivation	1.5–2	high-quality vegetable protein, source of vitamin B, potassium, phosphorus, magnesium, calcium, polyphenols, bioactive peptides, starch and dietary fiber	reduce the risk of cardiovascular diseases and diabetes, protective effect against various cancers, antioxidant, antihypertensive, anti-inflammatory and cholesterol-lowering effects, appetite-suppressing effect	Compared to animal protein sources, peas are more sustainable, requiring less land and water while producing lower greenhouse gas emissions, having the ability to fix nitrogen from the atmosphere, reducing the need for synthetic fertilizers and enhancing soil quality	antinutritional substances such as protease inhibitors, phytic acid, oxalates and tannins, undesirable beany aftertaste, and a complex spherical structure
[65,66,67,69,71,72,73,74,75,76,262]	Faba bean	1.36	20–35	Traditional cultivation	0.7–2	High protein content; rich in lysine; high-starch fraction; bioactive compounds; source of iron, phosphorus, magnesium, potassium, B vitamins; oligosaccharides	Possesses immune-modulating properties; lowers serum LDL and VLDL cholesterol levels; promotes the growth of beneficial gut microbiota; reduces chronic inflammation; lowers the risk of colon cancer	Fix atmospheric nitrogen; reduce the need for nitrogen fertilizers;	antinutritional factors such as vicine and convincing can trigger favism in individuals with glucose-6-phosphate dehydrogenase (G6PD) deficiency; Antinutritional factors such as lectins, saponins, trypsin inhibitors, phytic acid, and condensed tannins
[82,84,85,86,87,88,89,90,91,92,93,94,95,96,263,264,265,266]	Lupin	0.57	31–52	Traditional cultivation	7–12	Bioactive peptides; dietary fiber; high levels of oleic, linoleic, linolenic, palmitic and stearic acids; vitamins: thiamine, niacin, riboflavin, tocopherols and other micronutrients, e.g., carotenoids, iron, zinc and manganese; phenolic compounds; phytosterols; squalene; polyphenols	Antioxidative, anti-inflammatory, hypoglycemic, lipid profile-improving and hypotensive properties; reduced risk of colon cancer; positive effect on lipid profile and blood pressure	Ability to efficiently fix nitrogen from the atmosphere; reduced dependency on synthetic fertilizers; contribution to the process of carbon sequestration; helps mitigate climate change by capturing and storing carbon in the soil; exhibits an incredibly high tolerance to drought and frost, enabling diverse farming opportunities across the globe	The presence of antinutritional components such as lectins, protease inhibitors, condensed tannins, cyanogenic glycosides, saponins, alkaloids, phytic acid and selected oligosaccharides
[106,107,108,111,112,267,268,269,270]	Hemp seeds	0.73	20–25	Traditional cultivation	25–35	Easily digestible and amino acid-rich protein; linoleic acid, α-linolenic acid, and oleic acid; dietary fiber; minerals: potassium, phosphorus, magnesium, calcium, iron, zinc, and copper; phenolic compounds; bioactive peptides; carotenoids; tocopherols; phytosterols, lignan-amides and hydroxycinnamic acid	Antioxidative, antimicrobial, antihypertensive and cytomodulatory properties; reduced risk of CVD, cancers, rheumatoid arthritis, hypertension, inflammatory and autoimmune diseases	Requires minimal water and pesticide supply; reduces environmental pollution by absorbing large amounts of CO_2_ from the atmosphere; helps with mitigating climate change; low environmental footprint	Presence of antinutritional components such as phytic acid, tannins, cyanogenic glycosides, trypsin inhibitor and saponins; contains psychoactive compound—tetrahydrocannabinol (THC)
[119,121,124,126,127,133,271]	Microalgae	0	6–71	Cultivation in water	2–22	Omega-3 fatty acids (DHA, EPA), vitamins B12 and D2, carotenoids, sulfated polysaccharides, sulfolipids, astaxanthin	Antioxidant; cholesterol-lowering, anti-inflammatory, antihypertensive and anticancer properties; eye and cardiovascular health support	Small cultivation area; production can utilize by-products from agriculture and industry and absorb 10 to 50 times more CO_2_ than land plants	Accumulation of heavy metals—arsenic, cadmium, lead, mercury; limited consumer acceptance—slightly fishy smell
[137,141,142,143,145,272]	Macroalgae	0.7	10–47	Cultivation in water	1–3	Vitamins A, B12, C, β-carotene, pantothenate, folic acid, riboflavin, niacin, omega-3 fatty acids (DHA, EPA)	Prevention and control of T2D; reduction in postprandial glucose levels, HbA1c, and HOMA—IR; antioxidative properties; reduced risk of CVD	Autotrophic nutrition and rapid growth; do not need soil for cultivation; growth rate faster than land plants	Accumulation of heavy metals—arsenic, cadmium, lead, mercury
[152,155,273]	Water lentils (duckweed)	0.4	35–40	Cultivation in water	1–14	High protein content; phytosterols; starch; soluble sugars; cell wall components, amino acids; phenols, and tannins	Nutritional value similar to common foods and safe as supplements; low allergy risk	The fastest growing flowering plant in the world produces vast biomass for food, biofuels, and biogas	Contains calcium oxalate raphides, which may increase the risk of kidney stones
[159,160,162,164,166,167,274]	Insects	1.43–13.16	67–72	Terraria	1–57	Complete profile of essential amino acids; high in leucine; source of iron, zinc, copper, phosphorus and vitamins B2, B5, B7, and folic acid	High digestibility; less saturated fatty acids and more monounsaturated and polyunsaturated fats than meat; chitin, as a source of dietary fiber, may improve lipid profile	Requires much less water, generates significantly fewer greenhouse gases, and is more efficient at converting feed into protein than livestock farming	Low consumer acceptance; higher levels of toxic elements such as aluminum and lead
[178,179,180,186,275,276,277]	Microbial protein	1–2.23	30–65	Bioreactor, fermentation, genetic engineering	1–20	High protein content; high lysine and methionine content; good source of B vitamins and polyunsaturated fatty acids	Precise fermentation enables the production of animal-derived proteins, high quality protein	Minimal land requirements and efficient resource utilization; reduced environmental footprint	High nucleic acid content, which may increase the risk of gout
[190,191,192,194,195,202,203,204,241]	Mycoprotein	1.14–4.15	45	Fermentation, genetic engineering	13	Protein with high biological value; dietary fiber—β-glucan (up to 75%) and chitin; vitamin B12; folic acid; minerals: zinc, magnesium, calcium, and phosphorus; linoleic and linolenic acids	Improved appetite regulation through effects on metabolic and satiety hormones; positive effect on lipid profile and blood pressure; reduced risk of CVD; antihyperlipidemic, antioxidative and antimicrobial properties; stimulation of muscle protein synthesis	The environmental impact of mycoprotein production is still limited, and its environmental implications remain controversial	May potentially cause allergic and gastrointestinal symptoms such as anaphylaxis, hives, nausea, vomiting, and diarrhea; presence of mycotoxins
[207,210,278,279,280]	Cultured meat	1.69–22.1	76–90	Bioreactor	9–10	The biological equivalent of traditional meat	Easy digestible; collagen rich in essential amino acids; reduced risk of foodborne illness and antibiotic resistance development	Reduced cruelty to animals; fast production; small land requirements; reduced greenhouse gas emissions; production in a closed system without manure; 100% edible meat (in comparison to 5–25% from traditional livestock)	High production costs; lower protein and micronutrient content in comparison to traditional meat

d.w.—dry weight; CVD—cardiovascular disease; THC—tetrahydrocannabinol; DHA—docosahexaenoic acid; EPA—eicosapentaenoic acid; T2D—type 2 diabetes; HOMA—IR—Homeostatic Model Assessment—Insulin Resistance; Kg CO_2_e per kg—kilograms carbon dioxide equivalent per kilogram of the product.

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
