# Peer review of "Nutritional Quality, Safety and Environmental Benefits of Alternative Protein Sources—An Overview"

_nutrients, 2025, doi:10.3390/nu17071148_

Round 1
Reviewer 1 Report
Comments and Suggestions for Authors
nutrients-3520199-peer-review-v1
This narrative review analyzes the health and environmental effects of plant proteins, which the authors refer to as "alternative" proteins (alternative to what?). Additionally, the inclusion of insects in their analysis offers an interesting perspective. The review is well-written, with the table providing a useful summary, but it does not introduce new or particularly engaging content.
Several other similar papers exist, such as PMID: 36501146 published in this journal, PMID: 39767070 from another MPDI journal, the extensive PMID: 39631999, and the insightful PMID: 39360272. It would be beneficial to read these papers, discuss their findings, and explain what distinguishes this paper as innovative.
The rationale behind the choice of protein sources needs clarification. For example, pea protein is currently popular among bodybuilders. Furthermore, the review does not mention the EAT-Lancet project and its strengths and weaknesses.
Lastly, researching precision fermentation is recommended, as it represents a potential future method for protein production, including the extraction of proteins from air and synthesis by engineered yeasts in fermenters.
Author Response
Dear Reviewer,
Thank you very much for the time and work you have put in revising our paper. We really appreciate your comments and suggestion, as they allowed us to make amendments that improve the research quality of the manuscript. All of the changes that were introduced are marked in yellow in the revised version submitted.
Below, we would like to address the key points of the review:
- We extensively studied the mentioned papers. They are indeed useful and insightful, however, we did find some gaps that we wanted to address in our paper. Our aim was to prepare compendium of knowledge on protein sources, which are alternatives to animal-based products. We wanted to include and describe different types of products, their impact on environment and health, with special attention to nutritional aspects, as all authors are closely associated with dietetics and medicine and work in a clinical setting. To further improve our paper and introduce novelty, we decided to include the information about amino acid profile and how to balance it. Additionaly, the last chapter was extended to provide our clinical perspective, which was combined with deep ethical and environmental concerns.
- We agree that the section about legumes needed extension. Therefore, we decided to describe pulses as the whole group and to add additional chapters about plants, which (in our opinion) deserve special attention.
- We decided to discuss the planetary diet in brief in the last section of the article.
- A small section about precision fermentation was added.
Kind regards,
Authors
Reviewer 2 Report
Comments and Suggestions for Authors
This paper is a review article about alternative protein sources to meat. It is broad, and covers several issues of what is ultimately a massive topic.
The paper is readable and provides lots of information in an accessible manner, which is great for a review. While none of this information is new, nor is this likely the only review on the subject, the presentation is acceptable.
Legumes is a broad category, and I wonder if complete protein legumes like soy should be listed separately from incomplete protein sources like black beans or nuts and such.
Is is worth noting which proteins are complete and which are not [that is, which have all essential amino acids]. Soy is complete, but rice is not, and rice needs to be eaten with complementary protein sources.
Table 1 is good, except for the column "types of protein" that divides all the proteins into "animal" or "plant." Fungi and microbes are not plants. Please delete this column.
Some readers may want to see more information about the carbon footprint of these protein sources per unit protein, obtainable from life cycle assessment studies. Some of those are referenced, but could this information be added to the table as a category of, say, amount of carbon per gram or kg protein?
Throughout the paper, scientififc names must be in italics.
Author Response
Dear Reviewer,
Thank you very much for the time and work you have put in revising our paper. We really appreciate your comments and suggestion, as they allowed us to make amendments that improve the research quality of our manuscript. All of the changes that were introduced are marked in yellow in the revised version submitted. Our aim was to prepare compendium of knowledge on protein sources, which are alternatives to animal-based products. We wanted to include and describe different types of products, their impact on environment and health, with special attention to nutritional aspects, as all authors are closely associated with dietetics and medicine and work in a clinical setting. Additionaly, the last chapter was extended to provide our clinical perspective, which was combined with deep ethical and environmental concerns.
Below, we would like to address the key points of the review:
- We agree that the section about legumes needed alterations. Therefore, we decided to describe pulses as the whole group, and then to add additional chapters about plants, which (in our opinion) deserve special attention.
- We decided to include the information about amino acid profile and how to create balanced meals with complementary amino acid profiles.
- The section about rice was removed, as this plant alone is not a good source of protein and we were refering to rice protein isolate. It did not fit the scope of this review.
- A lot of amendments were made to Table 1 and we believe that those changes corresponds with your suggestions.
Kind regards,
Authors
Reviewer 3 Report
Comments and Suggestions for Authors
This review explores alternative protein sources that offer a more ethical and eco-friendly approach compared to conventional options. It discusses plant-based, aquatic, insect, microbial, and cell-cultured proteins, emphasizing their reduced environmental impact and higher ethical production standards. The article is comprehensive but requires structural cohesion (transitions, subheadings), logical clarity (explicit problem-solution links), and explanatory depth (quantitative data, definitions). Some further comments are as follows:
- There is a grammatical error in original introduction section: "...in the next 25 years, what will be associated with an estimated increase...","what" should be replaced with "which".
- Terms like "mycoprotein" and "precision fermentation" lack introductory definitions.
- Environmental benefits of insect farming (Section 5.1) lack quantitative comparisons.
- The discussion on overcoming sensory barriers (such as the fishy odor of algae) is rather limited and should be further supplemented.
- The link between environmental/ethical issues and specific protein solutions is implicit.
- The "future research" items listed in the conclusion section lack a hierarchical structure. It would be advisable to rank the critical research needs.
This review explores alternative protein sources that offer a more ethical and eco-friendly approach compared to conventional options. It discusses plant-based, aquatic, insect, microbial, and cell-cultured proteins, emphasizing their reduced environmental impact and higher ethical production standards. The article is comprehensive but requires structural cohesion (transitions, subheadings), logical clarity (explicit problem-solution links), and explanatory depth (quantitative data, definitions). Some further comments are as follows:
- There is a grammatical error in original introduction section: "...in the next 25 years, what will be associated with an estimated increase...","what" should be replaced with "which".
- Terms like "mycoprotein" and "precision fermentation" lack introductory definitions.
- Environmental benefits of insect farming (Section 5.1) lack quantitative comparisons.
- The discussion on overcoming sensory barriers (such as the fishy odor of algae) is rather limited and should be further supplemented.
- The link between environmental/ethical issues and specific protein solutions is implicit.
- The "future research" items listed in the conclusion section lack a hierarchical structure. It would be advisable to rank the critical research needs.
Author Response
Dear Reviewer,
Thank you very much for the time and work you have put in revising our paper. We really appreciate your comments and suggestion, as they allowed us to make amendments that improve the research quality of our manuscript. All of the changes that were introduced are marked in yellow in the revised version submitted.
Please, find our responses to your comments below:
- The error was corrected.
- We did introduce neccessary definitions.
- We decided to elaborate on environmental benefits of insect farming.
- We decided to elaborate on sensory barriers of aquatic sources.
- The last chapter was extended to provide our clinical perspective, which was combined with deep ethical and environmental concerns.
- Some amendments were made to the "Conclusion" section.
Kind regards,
Authors
Reviewer 4 Report
Comments and Suggestions for Authors
The authors tried to review various alternative protein resources. However, there are a few reviews on specific alternative protein (such as microalgae, insect protein). This review tried to summarize all of the alternative protein resources, which is difficult.
- The topic is too big in this review. Each subsection in this review can be expanded to an individual review.
- The review is more like a book chapter, which can provide a simple introduction of the alternative proteins and is more suitable for undergraduate students. It’s not qualified as an academic review.
- Tables or figures should also be added to make it easier for the readers to get your point. At this form, it’s difficult to get some effective information.
- Rice can’t be regarded as an alternative protein source, in my opinion.
- Table 1 should be expanded to add more representative studies in the last years.
- The main problem for this review is that I can’t get the opinion of the authors. They just summarized the previous literature. It’s also difficult to summarize this big topic.
Author Response
Dear Reviewer,
Thank you very much for the time and work you have put in revising our paper. We really appreciate your comments and suggestion.
We introcuded some changes to improve the quality of the manuscript. All of them are marked in yellow in the revised version submitted. Our aim was to prepare a compendium of knowledge on protein sources, which are alternatives to animal-based products. We wanted to include and describe different types of products, their impact on environment and health, with special attention to nutritional aspects, as all authors are closely associated with dietetics and medicine and work in a clinical setting. We decided to include the information about amino acid profile and how to create balanced meals with complementary amino acid profiles. Additionaly, the last chapter was extended to provide our clinical perspective, which was combined with deep ethical and environmental concerns.
The section about rice was removed, as this plant alone is not a good source of protein and we were refering to rice protein isolate. It did not fit the scope of this review. Moreover, some amendments were made to Table 1 to summarise the most important information about described protein sources.
Kind regards,
Authors
Reviewer 5 Report
Comments and Suggestions for Authors
The topic of this manuscript is motivated by the shortage of human protein resources, leading to the development of protein substitutes. Authors provide examples from the fields of plant proteins, insects, and microbial proteins, and discusses the application prospects from the perspectives of ethics, environment, and health. However, there is an issue that must be addressed: the manuscript focuses too much on the first half, while the latter half is underdeveloped. Here are my suggestions:
In Lines 26-27, the keyword "novel foods" seems inappropriate. It could be changed to "alternative protein sources."
In the introduction section (Lines 30-58), the author should add 50-100 words to describe the important role of protein in human dietary composition and health. For example, proteins are used for the synthesis of body tissues.
In Line 59, under "Plant-based protein sources," it could be noted that legume forage sprouts, such as alfalfa sprouts, are also high-protein raw materials with high nutritional value.
The description of rice as a protein substitute in Lines 154-179 is inappropriate. Rice already has a high global consumption and is primarily a source of energy. Its protein content and quality are relatively low, making it unsuitable as a substitute.
The reference citation format in Lines 265-267 does not comply with the journal's requirements. This issue should be corrected here and throughout the text.
What is the difference between "Mycoproteins" in Line 372 and "Microbial proteins" in Line 480?
The discussion of "Alternative protein sources – ethical, environmental and health-related aspects" in Lines 612-629 is too brief. The author should elaborate on each aspect separately: ethical, environmental, and health-related. This is the innovation point of the review, rather than just describing specific protein substitutes. The author needs to expand this section significantly, especially regarding the impact on human health.
The reference citation format in Line 662 is inconsistent. For example, some journals are cited in full, while others are abbreviated. In Lines 1060-1061, what is "IJMS" in reference 169? This is not a standard abbreviation for a journal. The author should also check similar issues, such as in Lines 828-829, where the journal for reference 68 should be "Journal of Animal Physiology and Animal Nutrition," not "Animal Physiology Nutrition" as currently written.
Author Response
Dear Reviewer,
Thank you very much for the time and work you have put in revising our paper. We really appreciate your comments and suggestion, as they allowed us to make amendments that improve the research quality of the manuscript. All of the changes that were introduced are marked in yellow in the revised version submitted.
Below, we would like to address your comments:
- The phrase "novel foods" was removed.
- Short protein characteristics was included in the Introduction.
- We included the information about the sprouts.
- The section about rice was removed, as this plant alone is not a good source of protein and we were refering to rice protein isolate. It did not fit the scope of this review.
- The citation format in lines 265-267 (according to the initial version) was corrected.
- Thank you for the important comment about mycoproteins. They were included in the microbial proteins section in the revised version of the manuscript.
- The last chapter was extended to provide our clinical perspective. It is now divided into three different sections: ethical, environmental and health-related.
- Citations format was revised.
Kind regards,
Authors
Round 2
Reviewer 1 Report
Comments and Suggestions for Authors
Good revision, but the authors dodged the comment on references that report similar conclusions. The result is that there is little new in this paper. Go back, re-read the suggested citations (there are more by the way) and frame this work in the larger picture.
Author Response
Dear Reviewer,
Apologies for missing this point. We went through the suggested papers and decided to add new paragraphs in the section 6.3 to address the issues discussed in them. Additionally, new section about methodology was added.
Thank you again for your time and effort that you put in revising our manuscript.
Kind regards,
Authors
Reviewer 4 Report
Comments and Suggestions for Authors
The authors have improved the manuscript and it can be accepted for publication.
Author Response
Dear Reviewer,
Thank you again for your time and effort that you put in revising our manuscript. Few additional changes were made to address all reviews received. They are marked in blue.
Kind regards,
Authors
Reviewer 5 Report
Comments and Suggestions for Authors
I have checked the revised version authors submitted, I confirmed that my previous concerns have been well addresssed. Thank you for your contribution to the protein resource, good luck and best wishes!
Author Response

(The authors gave the same response as above.)
